# Customizable FDM-based zebrafish larva mold for live imaging

**Marcela Xiomara Rivera Pineda[1,2,3,*], Jaakko Lehtimäki[1,2,3,*] and Guillaume Jacquemet[1,2,3,4,‡]**

## ABSTRACT

Accurate and reproducible orientation of zebrafish larvae is essential for high-resolution live imaging but remains difficult with standard agarose mounting. Although previous methods have used 3D-printed tools, often fabricated via stereolithography, here, we present an orientation tool made via fused deposition modeling and enhanced with a thin resin coating to improve surface smoothness and performance. This design creates consistent, larval-shaped wells that orient larvae dorsoventrally. The molds, designed to fit standard glass-bottom imaging dishes, are inexpensive to produce on common desktop 3D printers and can be easily customized to fit different imaging setups using the open-source design files. This approach enables reliable, long-term live imaging of multiple larvae and extends the use of low-cost additive manufacturing in zebrafish imaging workflows.

KEY WORDS: Zebrafish larvae, Live imaging, 3D printing, Fused deposition modeling

## INTRODUCTION

Zebrafish (*Danio rerio*) embryos and larvae are popular vertebrate model organisms for live imaging and tracking phenomena from single-cell behavior to tissue-scale morphogenesis. This is particularly due to their optical transparency and rapid *ex utero* development (Adhish and Manjubala, 2023). The large clutch sizes further enable simultaneous tracking and imaging of multiple age-matched siblings. Developing larvae lie on their side until the inflation of the swim bladder around 120 h post-fertilization (hpf) (Kimmel et al., 1995). These orientations can make imaging tissues located more dorsally, such as the brain, challenging. Therefore, immobilization techniques using low-melting-point agarose or specialized molds have been developed to maintain developing zebrafish embryos in the desired position (Geng and Peterson, 2021; Jutoy et al., 2025; Kleinhans and Lecaudey, 2019; Masselink et al., 2014; Miller et al., 2025; Peravali et al., 2011; Sim et al., 2025; Westhoff et al., 2013; Wittbrodt et al., 2014). Due to the temperature sensitivity of larvae, agarose needs to be administered closer to the gelling point, which complicates the successful immobilization of multiple larvae in a single imaging dish for their simultaneous live imaging. Recent advances in additive manufacturing, specifically 3D printing, have enabled the creation of custom orientation tools that can produce different types of embryo- and larvae-shaped wells in agarose and achieve the desired orientation for imaging (Jutoy et al., 2025; Kleinhans and Lecaudey, 2019; Masselink et al., 2014; Miller et al., 2025; Wittbrodt et al., 2014).

While most published 3D-printing designs rely on stereolithography (SLA) for high precision (Jutoy et al., 2025; Kleinhans and Lecaudey, 2019; Masselink et al., 2014; Miller et al., 2025), fused deposition modeling (FDM) offers a practical, accessible alternative. FDM and SLA-based 3D printing each have their own strengths and limitations. Abbasi et al. (2025) compared the two approaches in terms of performance and cost. SLA-based 3D printers can achieve very high resolution and smooth surfaces, enabling them to reproduce fine details with great accuracy. However, this comes with trade-offs, including the need for resin handling, solvent washing, post-curing under ultraviolet (UV) light, and proper disposal of chemical waste. Conversely, FDM-based 3D printers extrude melted thermoplastic filaments, such as polylactic acid (PLA). They are easier to operate and maintain, though their resolution is limited by the nozzle size and inherent material properties, resulting in less-detailed features. The two technologies also differ economically: entry-level SLA printers are generally affordable, but the ongoing costs for resin and consumables are higher. FDM printers may have a higher initial cost, but the materials are inexpensive and widely available. PLA filaments also offer a more environmentally friendly option than photopolymer resins, which produce hazardous waste. From a learning standpoint, most users find FDM easier to master, while SLA requires more attention to post-processing procedures. No single method is ideal for every situation, and both have their advantages depending on the priorities.

Here, we show that FDM is a practical, accessible method for creating molds for high-resolution live imaging of zebrafish larvae. Although it has a lower print resolution and slightly less-defined cavity geometry compared to SLA, the resulting wells consistently secure larvae in a dorsoventral position, enabling parallel, brain-focused imaging. Notably, the stability of this positioning allows immobilization with just 0.2% agarose, maintaining viability and enabling uninterrupted, high-quality time-lapse imaging for 24 h or more, resulting in rich, cellular-level datasets over extended periods.

## RESULTS

High-quality brain imaging of early zebrafish larvae requires positioning multiple specimens, preferably close to the coverslip surface, in a head-down orientation, which is difficult to achieve with standard agarose mounting techniques. Our first attempt at fabricating the zebrafish molds followed the approach of Strobel et al. (2018), using a 3D-printed negative to cast molds in polydimethylsiloxane (PDMS). Although these PDMS molds reproduced the larval cavity shape more faithfully than our final fully 3D-printed versions, they presented practical challenges: the PDMS often adhered to the

[1]Faculty of Science and Engineering, Cell Biology, Åbo Akademi University, 20520 Turku, Finland. [2]Turku Bioscience Centre, University of Turku and Åbo Akademi University, 20520 Turku, Finland. [3]InFLAMES Research Flagship Center, University of Turku and Åbo Akademi University, 20520 Turku, Finland. [4]Foundation for the Finnish Cancer Institute, Tukholmankatu 8, 00290 Helsinki, Finland.
*These authors contributed equally to this work

‡Author for correspondence (guillaume.jacquemet@abo.fi)

M.X.R.P., 0009-0000-9838-9533; J.L., 0000-0002-0002-0242; G.J., 0000-0002-9286-920X

Biology Open

agarose gel, causing the gel to detach from the imaging dish. This issue led us to move toward a fully 3D-printed mold design.

Most published 3D-printed orientation molds rely on SLA printing (Jutoy et al., 2025; Kleinhans and Lecaudey, 2019; Masselink et al., 2014; Miller et al., 2025), but we lacked access to this technology and aimed to avoid the demanding post-processing steps associated with 3D printing. Therefore, we developed an FDM-based immobilization strategy optimized for brain imaging. To design our molds, we drew inspiration from the molds proposed by Geng and Peterson (2021) and made some modifications to adapt them to our imaging setup. First, we introduced a circular mold format that fits standard imaging dishes with 14- and 21-mm coverslips. Both versions share the same internal slot geometry, extending fully to the glass surface. After iterative testing to improve performance, we adjusted two key slot dimensions: a longer tail and deeper slots. The tail length was increased to accommodate the size of the developmental stages we imaged, and the depth was increased to reduce surface contact between the mold and the agarose gel, helping prevent the gel from detaching from the dish. To further reduce irregularities caused by FDM printing, we applied a thin resin coating as a simple post-processing step, which enhanced mold performance without requiring complete SLA fabrication (Fig. 1A-C). Although our molds were designed in computer-aided design (CAD) with sharp, triangular profiles, the FDM printing resolution limit of 0.4 mm nozzle size resulted in rectangular features instead (Fig. 1D). To improve handling during and after agarose gelling, we added a

two-part, stamp-like design, with the upper part being easily attachable or detachable when needed (Fig. 1E).

With this custom orientation mold, zebrafish larvae at 50 hpf and older can be securely mounted in a dorsoventral, head-down position, requiring only 0.2% agarose for additional immobilization (Fig. 2A,B). To quantitatively assess the reproducibility of larval positioning, we mounted 52 hpf wild-type larvae on one small (ø 13 mm) and four large (ø 20 mm) molds and acquired upright brightfield images. Using left-right eye symmetry as a readout for consistent dorsoventral orientation relative to the imaging plane, we calculated similarity scores for each larva (area S 0.91±0.06), supporting consistent and reproducible positioning of larvae in the molds (Fig. S1). Additionally, in our experience, larvae confined in agarose molds do not show reduced viability for at least 24 h after mounting (Table S1).

Not only does our mold design improve the consistency of the desired dorsoventral positioning, but it also prevents larval rotation, enabling precise placement of the heads near the glass coverslip of the imaging dish by applying a gentle downward pressure with a gel-loading pipette tip during mounting. Using 54 hpf *Tg(fli1:GFP-CAAX)* (vascular endothelial cells, cyan) and *Tg(gata1a:DsRed)* (erythrocytes, red) positive larvae, we performed spinning-disk confocal imaging (with a 25× immersion objective) of vascular development at least up to 80 µm inside the developing brain without significant signal loss or compromised image quality during overnight acquisition (Fig. 2C). Additionally, as shown by the

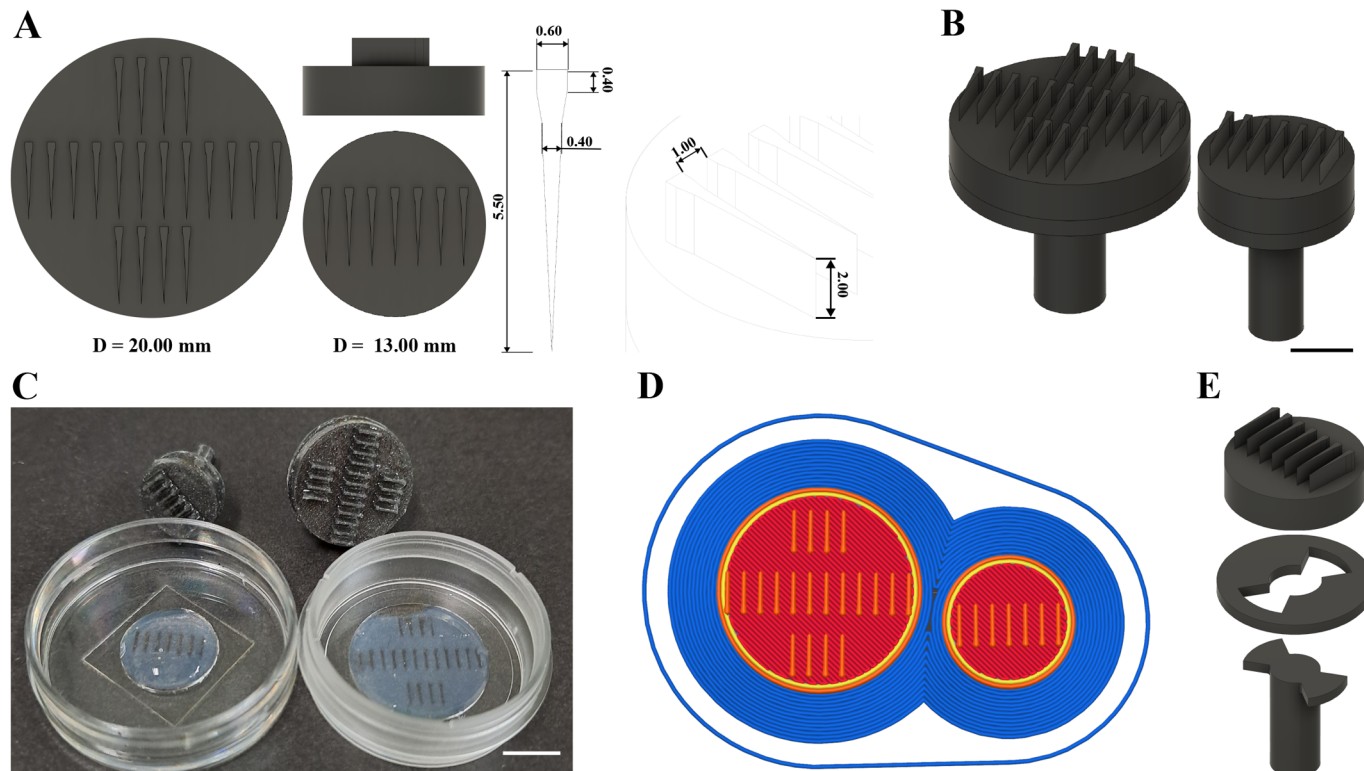

**Fig. 1. FDM-printed mold design.** (A) CAD renderings of two circular mold sizes (D/ø=20 and 13 mm) with slot dimensions in millimeters. Each slot measures 0.40 mm wide at the narrowest point and 0.60 mm at the widest and extends 5.50 mm deep; the spacing between slots is 1.00 mm, and the slot height above the base is 2.00 mm. The smaller mold was designed for use with 14 mm coverslips, while the larger was tested with 21 mm coverslips. The mold design was kept circular rather than rectangular to fit tightly within the well and stay level. Design files for both sizes are available for download via the associated GitHub repository. (B) Perspective 3D views of the assembled molds. Scale bar: 5 mm. (C) Wells in 1.5% low-melting-point agarose were made using the seven- and 20-tooth molds in 14 and 21 mm glass-bottom dishes, respectively. Scale bar: 10 mm. (D) PrusaSlicer G-code preview illustrating how FDM resolution (0.4 mm nozzle) converts triangular cavities into rectangular slots. (E) Exploded view of mold components, with a removable handle to facilitate positioning in small dishes.

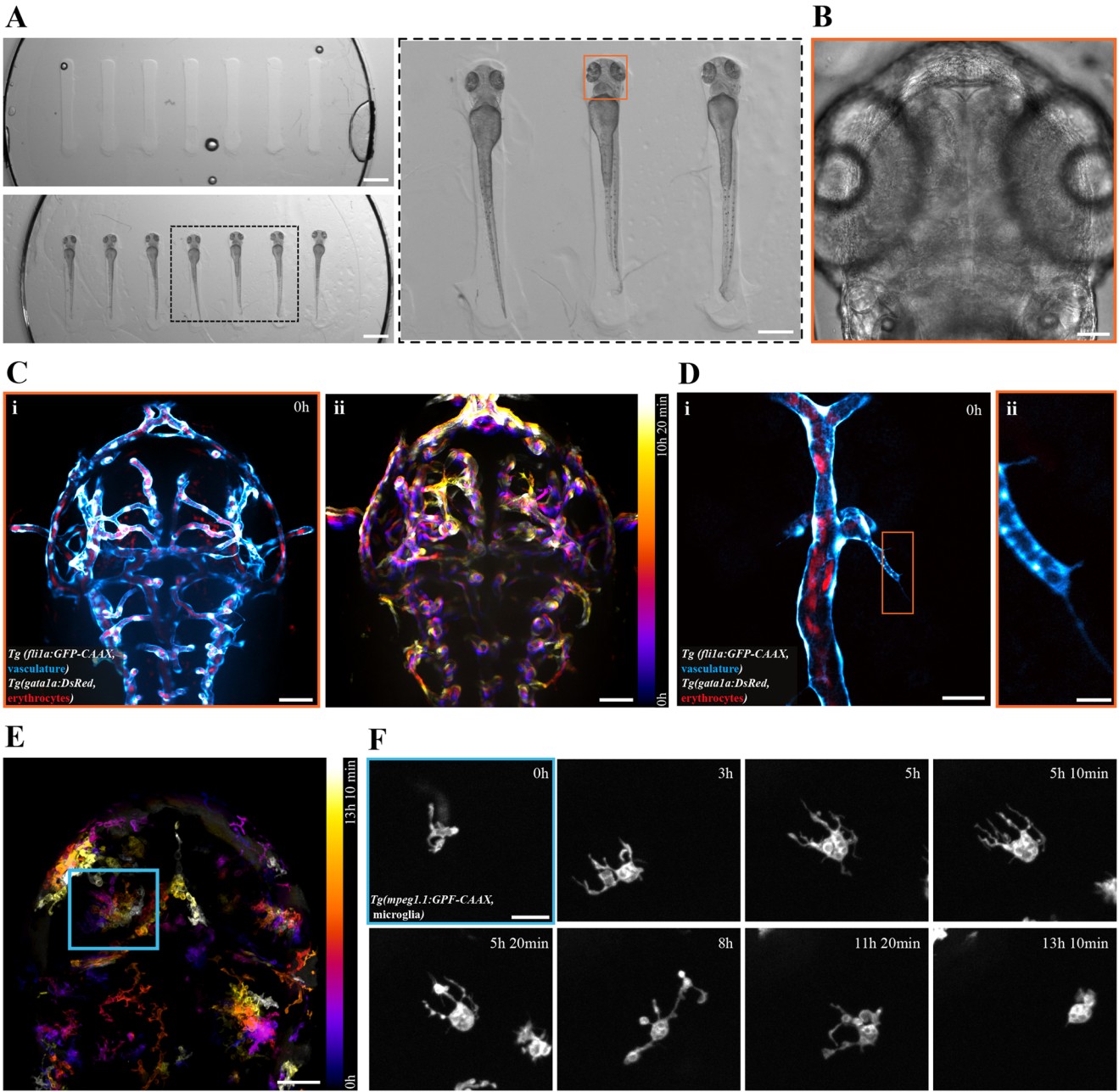

**Fig. 2. FDM printing-based mold design enables consistent live imaging of microglial motility and brain vascularization in developing zebrafish larvae.** (A) Top left: image showing empty agarose wells created using the seven-teeth mold. Bottom left: image showing 56 hpf *Tg(fli1a:GFP-CAAX)* and *Tg(gata1a:DsRed)* larvae inserted into the wells. The black-dashed box indicates the area magnified in the next panel. Right: magnified area of mounted larvae. Live brightfield imaging of the middle larvae, marked with an orange box, is presented in panel G. Scale bars: left panels 1 mm, right panel 500 μm. (B) Brightfield midplane-volume image of the 56 hpf *Tg(fli1a:GFP-CAAX)* and *Tg(gata1a:DsRed)* larvae from panel F (orange boxed region), before the start of time-lapse acquisition, shown in panels C and D (orange box). Scale bar: 50 μm. (C) Maximum intensity projection of horizontal sections from overnight time-lapse imaging of vasculature, *Tg(fli1a:GFP-CAAX)* (cyan), and erythrocytes, *Tg(gata1a:DsRed)* (red), in the developing brain of the mounted larvae (same larva as shown in panels A and B, orange box). The first time point is shown in panel i. Temporal color coding of the fli1:GFP signal over the acquisition (time lapse, 20-min imaging interval, 32 time points) (ii). The color bar indicates the transition from time points 1 (magenta) to 32 (bright yellow). See also Movie 1. Scale bars: 50 μm. (D) High-magnification (63×) imaging of the brain vascularization process. The maximum intensity projection of the first acquisition time point is shown in (i). The magnified area (indicated with an orange rectangle) shows filopodia-like protrusions extending from the fli1:GFP-positive (cyan) endothelial tip cell (ii). The *gata1a:DsRed*-positive erythrocytes (red) inside a more mature blood vessel are also shown. See also Movie 1. Scale bars: 25 and 5 μm, respectively. (E) Temporal color coding of microglial dynamics in *Tg(mpeg1.1:GFP-CAAX)* 72 hpf larvae over the acquisition period (10-min imaging interval, 80 time points). The color bar indicates the transition from time points 1 (magenta) to 80 (bright yellow). Scale bar: 50 μm. (F) Selected time points from panel E (blue box), showing microglial *Tg(mpeg1:GFP-CAAX)* (gray) migration and process elongation. See also Movie 2. Scale bar: 20 μm.

temporal projections over the acquisition period, the mold minimizes larval movement during imaging. After an overnight session, we continued to observe detailed vascular processes in the same larvae using a high-magnification, high-numerical aperture (NA) objective and a shorter sampling interval, which allowed us to visualize protrusions during vessel sprouting (Fig. 2D, Movie 1) with effective

lateral and axial resolutions (based on a voxel size of $0.1007 \times 0.1007 \times 0.5$ μm$^3$) of approximately 0.27 and 1.0 μm, respectively. To demonstrate the mold design's versatility in imaging different cell types in the brain, we also imaged microglial motility in 72 hpf *Tg(mpeg1.1:GFP-CAAX)* larvae (Fig. 2E). As with vascular sprouting, we captured the extension of dynamic microglial processes surveilling the surrounding tissue (Fig. 2F, Movie 2). At both 54 and 72 hpf, the mounted larvae exhibited normal development during imaging and maintained a healthy heartbeat at the end of the acquisition.

Our FDM-based mold design, enhanced with a thin resin coating and engineered for optimal brain imaging, provides a practical, reliable solution for long-term imaging of multiple zebrafish larvae at 50 hpf. To extend usability across developmental stages, we also provided a CAD-based redesign workflow where well dimensions (head width, neck width, and length) can be adjusted in a single sketch, allowing the mold to be adapted for both younger embryos and older larvae using published stage measurements (Kimmel et al., 1995) and our recommended 3D design dimension sets in the associated GitHub repository.

When transferred into agarose, the cavity dimensions help position larvae in a stable dorsoventral, head-down orientation while requiring only a very low agarose concentration, thus preserving physiological integrity during extended live-imaging sessions. Due to their low cost, ease of fabrication, and compatibility with standard dish formats, these molds offer an accessible option for improving the consistency and quality of high-resolution live imaging of zebrafish.

## DISCUSSION

We have developed an easy and affordable method to keep zebrafish larvae properly oriented during live imaging. Using a custom FDM-printed mold to create larvae-shaped wells in agarose, this technique ensures consistent sample positioning with minimal immobilization. The mold was made from low-cost PLA filament with a resin coating and can be easily reproduced or modified; all design files and instructions are openly available through the related GitHub repository associated with this paper. This approach builds on previous orientation methods using 3D-printed templates (Kleinhans and Lecaudey, 2019; Miller et al., 2025; Wittbrodt et al., 2014) and demonstrates that despite its trade-off in precision compared to SLA printing, the FDM-printed molds remain practical and accessible tools for ensuring consistent zebrafish larvae orientation while offering a lower learning curve, reduced consumable costs, and less post-processing and handling.

In practice, the tool allows imaging the brain development of multiple larvae from a more consistent dorsal plane. The time spent mounting specimens onto the mold is not significantly longer than the time needed to manually reposition similar numbers of larvae as the agarose solidifies. Our mold design can be used from 50 to 72 hpf and likely up to 5 days post-fertilization (dpf), though it was not directly tested here. In brain development, these stages are commonly used to study secondary neurogenesis (Chapouton and Bally-Cuif, 2004), microglial infiltration and homeostasis (Casano et al., 2016), or early neuronal activity (Randlett et al., 2015). Uniform head positioning enables tissue-level features of interest, such as the midbrain vascularization process, to appear at similar spatial and axial locations across horizontal optical sections and across different imaged larvae. For this end, potential head tilting toward the side, which is more likely with traditional agarose mounting techniques, complicates downstream visualization and analysis to separate the retinal vasculature from that of the midbrain (Gore et al., 2012). The same would hold for examining other brain-resident cells, such as microglia, that inhabit both the retina and other brain regions. The two mold designs presented here can also be applied to small-scale screening experiments that commonly use 96-well plate formats and offer limited control over larval orientation.

Overall, and agreeing with earlier reports on standardizing embryo and larvae orientation, we find that this tool improved the throughput and consistency of our imaging workflow (Geng and Peterson, 2021; Jutoy et al., 2025; Kleinhans and Lecaudey, 2019; Masselink et al., 2014; Miller et al., 2025; Peravali et al., 2011; Sim et al., 2025; Westhoff et al., 2013; Wittbrodt et al., 2014). The accessibility of desktop 3D printing allows most laboratories to create such tools, needing only a printer, suitable materials, and minimal post-processing. While zebrafish younger than 50 hpf cannot be imaged using this specific design due to the larger size of the yolk, which exceeds the diameter of the individual positions, we provide a step-by-step guide in the associated GitHub repository to modify the editable files and update the mold (tooth) dimensions (Fig. 1A) according to the stage of the larvae. The design can be scaled to accommodate both younger (<50 hpf) embryos and older larvae (up to 120 hpf), consistent with previous reports (Geng and Peterson, 2021; Kleinhans and Lecaudey, 2019; Miller et al., 2025). Additionally, the tooth dimensions can be adapted for other orientations (ventral or lateral) to optimize imaging of different tissue types.

In summary, the accessibility, adaptability, and performance of the FDM-based molds offer a practical solution for achieving consistent larval orientation and long-term live imaging, enabling standardized zebrafish mounting workflows. They also demonstrate that SLA is not the only viable option, and laboratories with existing FDM printers do not need to buy a resin printer solely for this purpose.

## MATERIALS AND METHODS
### Mold design and fabrication

The zebrafish molds were modeled in the commercial CAD software Fusion 360 (Autodesk, USA). The design is a modified version of the one proposed by Geng and Peterson (2021). The CAD files, in STL format, were converted to G-code using the open-source software PrusaSlicer v2.5.0 (Prusa Research, Czech Republic). The molds were printed on a desktop 3D printer (Prusa i3 MK3S+, Prusa Research, Czech Republic) using PLA (Prusament PLA Jet Black, Prusa Research, Czech Republic) on a textured, powder-coated print sheet (MD-28, Prusa Research, Czech Republic). All other components used the default settings. The molds were printed with the following G-code parameters: a print speed of 60 mm/s, nozzle temperature of 215°C, bed temperature of 60°C, layer height of 0.1 mm, and infill percentage of 5%. Each mold was designed to hold seven and 20 larvae, respectively, from 48 to 5 dpf. Printing settings are publicly available in the associated GitHub repository.

### Post-processing treatment

To minimize adhesion to the agarose gel, the printed molds were sanded with common nail files to remove rough edges and increase surface porosity for resin infiltration. A thin coat of two-part epoxy (Gédéo Crystal, Pébéo, 766150), mixed according to the manufacturer's instructions, was then applied over all surfaces of the mold that contact the agarose gel. A minimum of 24 h was allowed for the resin to cure fully.

### Zebrafish husbandry

*Tg(mpeg1.1:GFP-CAAX)*, *Tg(fli1a:GFP-CAAX)*, and *Tg(gata1a:DsRed)* lines were maintained and incrossed or outcrossed at 28.5°C, with larvae screened at 48 hpf (staging according to Kimmel et al., 1995) for the presence of transgenes. Adult zebrafish were housed and used for breeding following Directive 2010/63/EU and license MMM/465/712-93 (Ministry

of Forestry and Agriculture). Embryos were grown in E3 medium supplemented with 0.2 mM 1-phenyl-2-thiourea (PTU, Thermo Fisher Scientific, 10107703) to prevent pigmentation. For larvae mounting and live imaging, E3 medium with PTU was further supplemented with 0.1 mg/ml tricaine methane sulfonate (MS-222, Sigma, E10521) to anesthetize the larvae. Live imaging was conducted on larvae between 50 and 85 hpf. Zebrafish younger than 5 dpf used in experiments are not classified as protected animals under Directive 2010/63/EU.

### Agarose well preparation

Glass-bottom 14 mm dishes (MatTek, P35G-0.170-14-C), 21 mm μ-Dishes (ibidi, 81158), and 28 mm glass-bottom dishes (Cellvis, D35-28-1.5-N) were used. The 28 mm dishes were used with a 20-well mold for live imaging of the microglial reporter line. Then, 950 μl of 1.5% low-melting-point agarose (Sigma, A9414) in E3 medium was heated to 70°C and cooled to 40°C, and 50 μl of MS-222 was added and thoroughly mixed. From this mixture, 250 and 400 μl were added to plates with 14 and 21 mm embedded glass coverslips, respectively, while 600 μl was used for the 28 mm dishes. The molds were carefully placed on top, ensuring no air bubbles were trapped between the mold and the agarose. The agarose was allowed to solidify at room temperature for 20 min. Afterward, the molds were removed by first gently lifting each edge and then the middle. The resulting wells (Fig. 1C) were washed with the E3 medium and used immediately for larvae mounting.

### Larvae mounting

We prepared 0.4% agarose by mixing 400 μl of 1% low-melting-point agarose (at 70°C), 550 μl of E3 medium, and 50 μl of MS-222 (4 mg/ml) and then incubating at 40°C. Larvae at 56 or 72 hpf were anesthetized with MS-222 in E3 medium for 5 min. They were then transferred to the same E3 medium on top of the wells and gently positioned using gel-loading tips (Thermo Fisher Scientific, 10411193). Excess medium was aspirated, and a 0.4% agarose solution was added dropwise to fill the remaining volume of each well. Because small amounts of E3 remain in the molds, the final agarose concentration is expected to be between 0.2% and 0.3%. The mounting process took approximately 5 min for the seven-position molds and about 15 min for the 20-position molds. The agarose was allowed to solidify at room temperature for 15 min. Afterward, the plates were placed into the microscope's imaging chamber, and 28.5°C E3 medium with 0.1 mg/ml MS-222 and 0.2 mM PTU was gently added to cover the wells.

### Imaging

Imaging of the mounted larvae was performed using a Marianas CSU-W1 (Intelligent Imaging Innovations, 3I) spinning-disk confocal microscope equipped with a Hamamatsu sCMOS Orca Flash 4.0 camera, an environmental chamber, and control (Okolab), set to 28.5°C. The system was operated via Zeiss Axio Observer 7 frame and Slidebook 6 software. Overnight time-lapse imaging used a 25×/0.8 multi-immersion objective (LD LCI Plan-ApoC, Zeiss, 420852-9871-799) at 20-min intervals, acquiring an 82 μm stack with a 2 μm step size. High-magnification time-lapse imaging employed a 63×/1.15 water-immersion objective (LD-C-Apo, Zeiss, 421887-9970) with a 2-min interval, capturing a 10 μm stack with a 0.5 μm step size.

For microglial imaging, an Eclipse Ti2-E microscope with X-Light V3 HTDS (Nikon/CrestOptics) was used, equipped with a Kinetix sCMOS camera (Photometrics) and a Dark Enclosure WarmBox, maintained at 28.5°C. Overnight time-lapse imaging was performed using a 25×/1.05 silicone objective (CFI Plan Apochromat Lambda S 25XC Sil, Nikon) at 10-min intervals, capturing an 80 μm stack with a 1.5 μm step size. Data were collected via the NIS-Elements AR software.

### Quantification of positioning reproducibility

To quantitatively assess the reproducibility of dorsoventral larval positioning in the molds, 52 hpf wild-type larvae ($n$=71) were imaged using upright brightfield microscopy (Zeiss Axio Zoom.V16, 16×). For each larva, the left and right eyes were manually outlined in Fiji (Schindelin et al., 2012) using freehand region of interests (ROI). ROI area measurements were obtained for each eye. To summarize left-right symmetry, a similarity score was calculated for each metric as $S=\min(L, R)/\max(L, R)$, where L and R are the measurements from the left and right eyes,

respectively, and S=1 indicates identical values. Similarity scores were calculated for all larvae, summarized as mean±SD, and the distributions were visualized as boxplots. All measurements were performed on single brightfield images acquired under identical illumination and magnification settings.

### Manuscript preparation

Figures were prepared using Fiji (Schindelin et al., 2012) and Inkscape. Movies were prepared with Fiji and compiled with OpenShot Video Editor (OpenShot Studios, LLC). GPT-5 (OpenAI) and Grammarly (Grammarly, Inc.) were employed as writing aids during manuscript preparation. The author also edited and validated all text sections. GPT-5 did not provide references.

### Acknowledgements

Imaging was performed at the Advanced Imaging Core, Turku Bioscience Centre, which is supported by the Finnish Advanced Microscopy Node of Euro-BioImaging Finland (Turku, Finland) and Turku Bioimaging. 3D printing was conducted at the Laboratory of Biophysics with access to facilities provided by Euro-BioImaging ERIC (Turku, Finland). The zebrafish work was conducted at the Zebrafish Core of Turku Bioscience Centre (University of Turku and Åbo Akademi University), partially supported by Biocenter Finland.

### Competing interests

The authors declare no competing or financial interests.

### Author contributions

Conceptualization: M.X.R.P., J.L.; Data curation: M.X.R.P., J.L.; Formal analysis: M.X.R.P., J.L.; Funding acquisition: G.J.; Investigation: M.X.R.P., J.L.; Methodology: M.X.R.P., J.L.; Project administration: G.J.; Resources: G.J.; Supervision: G.J.; Validation: M.X.R.P., J.L.; Visualization: M.X.R.P., J.L., G.J.; Writing – original draft: M.X.R.P., J.L., G.J.; Writing – review & editing: M.X.R.P., J.L., G.J.

### Funding

This study was funded by the Research Council of Finland (338537, 371287, and 374180 to G.J.), the Sigrid Jusélius Foundation (to G.J.), the Cancer Society of Finland (Syöpäjärjestöt; to G.J.), and the Solutions for Health strategic funding for Åbo Akademi University (to G.J.). Additionally, this research was supported by the InFLAMES Flagships Programme of the Research Council of Finland (decision numbers: 337530, 337531, 357910, and 357911). G.J. is supported by the Finnish Cancer Institute (K. Albin Johansson Professorship). This work was supported by the Research Council of Finland, FIRI 2023 (grant decision numbers: 359073 and 358879) and FIRI 2024 (grant decision numbers: 367582 and 367577). Open Access funding provided by the University of Turku. Deposited in PMC for immediate release.

### Data and resource availability

All design files, including technical drawings, Fusion 360 project files (.f3d), neutral CAD files (.step), and mesh files (.stl), as well as printing settings, both PrusaSlicer files (.ini and 0.3mf) and complete parameter configurations for reproduction in other slicers and finalized G-code, are available in the associated GitHub repository (https://github.com/CellMigrationLab/zebrafish-FDM-molds) archived in Zenodo (https://doi.org/10.5281/zenodo.18713363). All relevant data and details of resources can be found within the article and its supplementary information.

### Peer review history

The peer review history is available online at https://journals.biologists.com/bio/lookup/doi/10.1242/bio.062406.reviewer-comments.pdf.

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
