## [Peer Review File · Biology Open]

Customizable FDM-based zebrafish larvae mold for live imaging

Marcela Xiomara Rivera Pineda, Jaakko Lehtimäki and Guillaume Jacquemet

DOI: 10.1242/bio.062406

Editor: Daniel Gorelick

Review timeline

Original submission:	3 December 2025
Editorial decision:	11 December 2025
First revision received:	24 February 2026
Editorial decision:	1 March 2026
Second revision received:	2 March 2026
Accepted:	4 March 2026

Original submission

First decision letter

MS ID#: bio.062406

MS Title: Customizable FDM-based zebrafish larvae mold for live imaging

Authors: Marcela Xiomara Rivera Pineda, Jaakko Lehtimäki and Guillaume Jacquemet

I have now reached a decision on the above manuscript.

The reviewer reports are shown at the bottom of this email.

As you will see, the reviewers gave favourable reports, but raised some critical points that will require amendments to your manuscript. I hope that you will be able to carry these out, because we would like to be able to accept your paper.

Regarding the comments of reviewer 1, please focus especially on comments 3 and 4. For reviewer

2, please address all of the major and minor comments except you don't need to address the following major comments (this means you can ignore them) unless you feel they will improve the manuscript: 'A lengthy introduction'; 'Figure arrangements'. In the revised manuscript, references should be formatted according to Biology Open's format - see below for more details. I should be grateful if you would also provide a point-by-point response detailing how you have dealt with the points raised by the reviewers in the 'Response to Reviewers' box. Please attend to all of the reviewers' comments, except for the two from reviewer 2 described above. If you do not agree with any of their criticisms or suggestions please explain clearly why this is so.

Reviewer 1

Comments for the author

Pineda et al describe a 3D printed mold for patterning agarose to enable mounting and imaging of zebrafish larvae. Although several studies have done something similar in the past, the novelty here is the use of 3D printing the mold using FDM (fused deposition modeling) instead of stereolithography (SLA). The primary benefit of using FDM over PLA is its lower cost and greater
© 2026. Published by The Company of Biologists under the terms of the Creative Commons Attribution License (<https://creativecommons.org/licenses/by/4.0/>).

ease of use. Overall, the manuscript is clear, and its strengths and weaknesses acknowledged. I only have minor comments to improve the paper.

1) The author's refer to their method as a way to image zebrafish embryos. However, they state (and show) that it only works for animals at least 50hpf. At this point the fish have hatched, and I would not call them embryos but larvae. I think this language needs to be fixed throughout the manuscript, and especially in the title. When I first read it I thought they had a way of imaging embryos from fertilization to ~40hpf. It is misleading to use the term embryos for the developmental stage they are working with. Use of 'larvae' is also consistent with the accepted nomenclature of Kimmel et al 1995 cited in the paper.

2) A few sentences in the introduction do not quite make sense to me:

a) Lines 40-42, it's described that 'anesthetized embryos' lie on their side until inflation of the swim bladder. It's not clear why 'anesthetized' is used here. It sounds like the authors are just describing normal development in the absence of manipulation.

b) Lines 43-45, How does the size of the yolk define the position of the embryo proper? This sentence does not make clear sense to me and it does not fit into the flow of ideas here. Why does this make imaging the brain difficult? I think what the authors are trying to say is that the position of the brain relative to the yolk makes it difficult to image at these early stages. But that is not what is being conveyed. It's also not clear why this is being mentioned at all given that the method presented would not help with this issue since the authors only use fish over 50hpf where this is less of an issue.

3) The novelty of this paper is the use of FDM vs SLA for 3D printing. The primary benefit of FDM is stated to be cost and ease of use. However, there are many online companies that will print items and ship them to you. What would be the benefit of purchasing a 3D printer yourself and printing the items instead of outsourcing this?

4) There is no mention of how the method developed affects the speed and ease of mounting larvae. How quickly can this be done? How long does it take to mount, for example, 10 larvae, for imaging using this method versus traditional approaches?

5) The fact that the novelty of the study lies in the use of FDM versus SLA based 3D mold printing needs to be stated more clearly in the abstract. Especially since this concept has already been published several times before.

6) The formatting of the references is incorrect. Last names should be listed before first names to make the references easier to navigate.

Reviewer 2

Comments for the author

The authors present a fused-deposition-modeling (FDM) approach to fabricate orientation molds for zebrafish early developing embryos that can be utilized with standard glass-bottom imaging dishes. The design holds embryos in a dorsal-ventral, head-down position with only 0.2% low-melting-point agarose, thereby preserving viability and enabling long-term time-lapse imaging of vascular development. Additionally, all design files and printing parameters are made publicly available via a GitHub repository. Overall, the work addresses a common bottleneck in zebrafish live imaging standardizing embryo orientation while offering a cost-effective, open-source alternative to resin-based (SLA) printing.

The manuscript presents a well-executed, low-cost solution for a pervasive technical challenge in zebrafish imaging. The methodology is reproducible, the data are convincing, and the open-source files will be valuable to a broad audience. However, the manuscript would benefit from additional quantitative validation, additional data for its application and minor clarifications. I therefore recommend minor revision (if the authors can address the quantitative reproducibility, viability, and statistical concerns) before acceptance.

Following, I have outlined the major and minor points that I believe would enhance the manuscript's quality and increase its chances of acceptance for publication.

Major Points

A lengthy introduction

The introduction is excessively lengthy due to the extensive comparison of FDM and SLA-based methods, which includes a detailed analysis of their similarities, differences, advantages, and disadvantages. This comparison is not suitable for the introductory section and most of the 2nd paragraph need to move to the "Discussion" section.

Scope of developmental stages

The current design is limited to embryos ≥ 50 hpf because of yolk size (Fig. 1A). The authors state that the tooth dimensions can be modified to accommodate younger stages, but no practical guide or example is provided. Including a brief "how-to" for redesigning the tooth geometry (e.g., STL modification steps) would greatly enhance the utility for the community.

Figure arrangement

The presence of FDM-printing-based mold design and vascularization in developing zebrafish embryos within the same figure may cause reader discomfort. Therefore, I recommend that the author separate the designing part and results into two distinct images.

Quantitative assessment of positioning reproducibility

While qualitative images demonstrate consistent orientation, a quantitative metric (e.g., angle variance of the head relative to the coverslip across embryos) would strengthen the claim of improved throughput and consistency. If not, it is better to discuss why you couldn't do the quantitative analysis (in the "Discussion").

Long-term viability assay

The manuscript mentions that larvae develop normally after release from agarose, but no data are shown (e.g., survival rate, heart rate, or behavioral assay after imaging). Providing at least a short viability assessment would reassure readers that the low agarose concentration and resin coating do not adversely affect development.

Statistical analysis of imaging quality

Figures showing signal-to-noise ratio (SNR) or resolution metrics for low- versus high-NA objectives are presented, but statistical tests (e.g., t-test) are not reported. Adding appropriate statistical analysis and p-values will substantiate the claim of "no significant loss of signal".

Application of the current technique

If the authors can provide additional results that suggest the current method can be applied to study neuronal development, regeneration, and other related processes, it would enhance the manuscript's impact. Therefore, if it is feasible to perform ablation of a specific neuronal population and capture time-lapse images of neuronal regeneration or fin clipping in transgenic fish, followed by imaging of regeneration, such data would be highly valuable.

Minor Points

Figure 1 B-C

Include scale bars for the printed molds.

Ensure gene/transgenic line names are italicized (e.g., *Tg(fli1a:GFP)*).

References

Verify that all cited works (e.g., Geng & Peterson 2021) are present in the bibliography and follow the journal's style.

Reviewer's Responses to Questions

Experimental quality

Does each figure have the proper controls?

If 'No', please indicate reasons in Comments for Author box below.

Reviewer #1:

- No

Reviewer #2:

- Yes
-

Were the data analyzed using appropriate statistical tests?

If 'No', please indicate reasons in Comments for Author box below.

Reviewer #1:

- No

Reviewer #2:

- No
-

Reproducibility

Were experiments performed using adequate number of biological replicates?

If 'No', please indicate reasons in Comments for Author box below.

Reviewer #1:

- No

Reviewer #2:

- Yes
-

Does the methods section provide sufficient detail to permit reproducibility?

If 'No', please indicate reasons in Comments for Author box below.

Reviewer #1:

- Yes

Reviewer #2:

- Yes
-

Completeness

Are the manuscript's conclusions supported by the data?

If 'No', please indicate reasons in Comments for Author box below.

Reviewer #1:

- Yes

Reviewer #2:

- No
-

Scholarship

Do the authors cite and discuss the merits of data that would argue for and against their conclusion?

If 'No', please indicate reasons in Comments for Author box below.

Reviewer #1:

- No

Reviewer #2:

- Yes
-

Does the manuscript title & abstract accurately reflect the contents of the manuscript, without hyperbole?

If 'No', please indicate reasons in Comments for Author box below.

Reviewer #1:

- Yes

Reviewer #2:

- Yes

First revision

Author response to reviewers' comments

Comments from the Reviewers:

Reviewer 1: Pineda et al describe a 3D printed mold for patterning agarose to enable mounting and imaging of zebrafish larvae. Although several studies have done something similar in the past, the novelty here is the use of 3D printing the mold using FDM (fused deposition modeling) instead of stereolithography (SLA). The primary benefit of using FDM over PLA is its lower cost and greater ease of use. Overall, the manuscript is clear, and its strengths and weaknesses acknowledged. I only have minor comments to improve the paper.

We would like to thank the reviewer for taking the time to read our work and propose valuable suggestions.

1) The author's refer to their method as a way to image zebrafish embryos. However, they state (and show) that it only works for animals at least 50hpf. At this point the fish have hatched, and I would not call them embryos but larvae. I think this language needs to be fixed throughout the manuscript, and especially in the title. When I first read it I thought they had a way of imaging embryos from fertilization to ~40hpf. It is misleading to use the term embryos for the developmental stage they are working with. Use of 'larvae' is also consistent with the accepted nomenclature of Kimmel et al 1995 cited in the paper.

We thank the reviewer for pointing out this mistake and inconsistency. The nomenclature has now been changed throughout the manuscript to better reflect the developmental time points used in our study.

2) A few sentences in the introduction do not quite make sense to me:

a) Lines 40-42, it's described that 'anesthetized embryos' lie on their side until inflation of the swim bladder. It's not clear why 'anesthetized' is used here. It sounds like the authors are just describing normal development in the absence of manipulation.

b) Lines 43-45, How does the size of the yolk define the position of the embryo proper? This sentence does not make clear sense to me and it does not fit into the flow of ideas here. Why does this make imaging the brain difficult? I think what the authors are trying to say is that the position of the brain relative to the yolk makes it difficult to image at these early stages. But that is not what is being conveyed. It's also not clear why this is being mentioned at all given that the method presented would not help with this issue since the authors only use fish over 50hpf where this is less of an issue.

We agree with the reviewer that the anesthesia does not provide any additional information regarding the sentence on developing larval orientation, and we have now updated the text accordingly. We also concur that the yolk size was included to better describe the challenges of imaging brain development in earlier developmental stages. Additionally, we have added mold designs suitable for earlier embryo stages to the revised manuscript, although the main focus remains on the larval stages. We have revised the text accordingly (lines 30-32).

“Developing larvae lie on their side until the inflation of the swim bladder around 120 hours post fertilization (hpf) (Kimmel et al., 1995). These orientations can make imaging of tissues with a more dorsal location, such as the brain, challenging.”

3) The novelty of this paper is the use of FDM vs SLA for 3D printing. The primary benefit of FDM is stated to be cost and ease of use. However, there are many online companies that will print items and ship them to you. What would be the benefit of purchasing a 3D printer yourself and printing the items instead of outsourcing this?

We appreciate the reviewer raising this point and agree that clarifying the practical reasons a laboratory might choose to print molds with an existing FDM printer instead of outsourcing is important. We want to emphasize that we do not suggest users purchase a 3D printer specifically for this purpose. Instead, we provide an FDM-based option for laboratories and research groups that already have access to FDM printers but not SLA printers, and we acknowledge that commercial 3D printing services remain a valid alternative. Additionally, we point out that this recommendation is already included at the end of the Discussion section, where we emphasize that *“laboratories with existing FDM printers do not need to buy a resin printer solely for this purpose.”*

4) There is no mention of how the method developed affects the speed and ease of mounting larvae. How quickly can this be done? How long does it take to mount, for example, 10 larvae, for imaging using this method versus traditional approaches?

We appreciate the reviewer for highlighting the importance of including this practical information in the manuscript. Excluding the mold casting time (20 minutes, as described in the methods), positioning the larvae takes approximately 5 minutes, and it takes about 15 minutes for the 7-position molds and 20 minutes for the 20-position molds. This includes the initial placement and fine adjustments after adding the thin agarose layer. Afterward, we allow an additional 15 minutes for the agarose to solidify before starting imaging.

Compared to simply placing the larvae on melted agarose and constantly repositioning them until it starts to solidify, the mold approach takes only about 20-30 minutes longer. A significant part of this time is spent casting the mold. To achieve uniform dorsoventral mounting of 20 larvae using our mold, traditional methods would require multiple mounting steps to obtain similar positioning. This is because manual repositioning becomes increasingly difficult as agarose solidifies, especially with larger numbers of larvae.

We have now included these details in the Discussion (lines 173-174)

“The time spent mounting specimens onto the mold is not significantly longer than the time needed to manually reposition similar numbers of larvae as the agarose solidifies.”

and updated these to the methods section of the manuscript (lines 251-252).

“The mounting process took approximately 5 minutes for the 7-position molds and about 15 minutes for the 20-position molds.”

5) The fact that the novelty of the study lies in the use of FDM versus SLA based 3D mold printing needs to be stated more clearly in the abstract. Especially since this concept has already been published several times before.

We thank the reviewer for this comment and agree that the abstract should clarify how our work relates to previous 3D-printed orientation tools. Therefore, we have revised the abstract to explicitly state that we present an FDM-based alternative for manufacturing zebrafish larvae molds.

“Accurate and reproducible orientation of zebrafish larvae is essential for high-resolution live imaging but remains difficult with standard agarose mounting. Although previous methods have used 3D-printed tools, often fabricated via stereolithography (SLA), here we present an

orientation tool made via fused deposition modeling (FDM) and enhanced with a thin resin coating to improve surface smoothness and performance. This design creates consistent, larval-shaped wells that orient larvae dorsoventrally. The molds, designed to fit standard glass-bottom imaging dishes, are inexpensive to produce on common desktop 3D printers and can be easily customized to fit different imaging setups using the open-source design files. This approach enables reliable, long-term live imaging of multiple larvae and extends the use of low-cost additive manufacturing in zebrafish imaging workflows.”

6) The formatting of the references is incorrect. Last names should be listed before first names to make the references easier to navigate.

We apologize for this mistake. The formatting has now been corrected in the revised manuscript.

Reviewer 2: The authors present a fused-deposition-modeling (FDM) approach to fabricate orientation molds for zebrafish early developing embryos that can be utilized with standard glass-bottom imaging dishes. The design holds embryos in a dorsal-ventral, head-down position with only 0.2% low-melting-point agarose, thereby preserving viability and enabling long-term time-lapse imaging of vascular development. Additionally, all design files and printing parameters are made publicly available via a GitHub repository. Overall, the work addresses a common bottleneck in zebrafish live imaging standardizing embryo orientation while offering a cost-effective, open-source alternative to resin-based (SLA) printing.

We thank the reviewer for taking the time to review our manuscript and for their excellent suggestions.

The manuscript presents a well-executed, low-cost solution for a pervasive technical challenge in zebrafish imaging. The methodology is reproducible, the data are convincing, and the open-source files will be valuable to a broad audience. However, the manuscript would benefit from additional quantitative validation, additional data for its application and minor clarifications. I therefore recommend minor revision (if the authors can address the quantitative reproducibility, viability, and statistical concerns) before acceptance. Following, I have outlined the major and minor points that I believe would enhance the manuscript's quality and increase its chances of acceptance for publication.

Major Points

A lengthy introduction

The introduction is excessively lengthy due to the extensive comparison of FDM and SLA-based methods, which includes a detailed analysis of their similarities, differences, advantages, and disadvantages. This comparison is not suitable for the introductory section and most of the 2nd paragraph need to move to the "Discussion" section.

We appreciate the reviewer's feedback on trimming and reorganizing the introduction. However, following the editorial advice, we have decided to keep the introduction as it is for now.

Scope of developmental stages

The current design is limited to embryos ≥ 50 hpf because of yolk size (Fig. 1A). The authors state that the tooth dimensions can be modified to accommodate younger stages, but no practical guide or example is provided. Including a brief "how-to" for redesigning the tooth geometry (e.g., STL modification steps) would greatly enhance the utility for the community.

We thank the reviewer for this suggestion and agree that providing a practical example of adapting the well (tooth) geometry would enhance the tool's usefulness for the community, especially during earlier developmental stages. To address this, we extracted stage-specific embryo diameters from Kimmel et al. (1995) (dorsoventrally positioned 28, 33, 42, 60, 72, and 120 hpf embryos and larvae, respectively) and used them to expand the mold's customization

throughout zebrafish development. Additionally, we have included a brief step-by-step “how-to” guide on modifying the well geometry. This material has been added to the project GitHub page (<https://github.com/CellMigrationLab/zebrafish-FDM-molds/blob/main/how-to/README.md>) and archived on Zenodo (<https://zenodo.org/records/18713363>) to facilitate easy redesign and reuse for users who wish to adapt the molds for different stages.

Text and figure added to GitHub:

How to redesign the wells:

Note 1: Changing one dimension of the first well propagates to the others. Therefore, after edits, you should always check for overlapping and spacing. *Note 2:* These steps can be used to modify other dimensions of the mold, such as the depth of the wells or the mold diameter. *Note 3:* All redesign steps were performed in Autodesk Fusion 360. Although Fusion 360 is subscription-based, a free educational license is available for academic users.

1. Open the corresponding mold file (.f3d file) in Fusion 360. Open the wells sketch by double-clicking the first sketch in the Timeline (highlighted by the red rectangle in Figure 3A).
2. Modify well dimensions by double-clicking the desired dimension as shown in Figure 3B (red rectangle). Edit the relevant dimensions (head width, neck width, length) according to the embryo/larva stage. See Figure 3E for suggested dimensions based on embryo stage (based on Kimmel et al. (1995)).
3. After resizing the wells, check the distance between the first well and the edge reference. Update it so the modified wells still fit cleanly within the diameter boundary. E.g.: 1.4 mm (Figure 3B, red triangle) to 1.2 mm (Figure 3C, red triangle) after changing the well geometry.
4. Verify spacing and pattern integrity by ensuring that the wells don't overlap after the update. If the wells overlap, modify the spacing by zooming in, double-clicking the Rectangular Pattern Constraint (dotted rectangle in Figure 3D), and adjusting the number of wells (green triangle in Figure 3D) and the distance between them (magenta triangle in Figure 3D).
5. Click Finish Sketch. In the Timeline, confirm that the extrude and cut features regenerated correctly.
6. Export the updated mold as an STL file and proceed to slicing and 3D printing. To export, go to File> Export..., and change the Type to STL Files (*.stl).

Figure 1. How to redesign the zebrafish molds.

(A) Autodesk Fusion 360 project file showing the mold with replicated wells arranged to fit a

standard glass-bottom imaging dish. The red rectangle highlights the sketch to edit to modify the wells' dimensions.

(B-C) The well sketch can be edited by opening the first sketch in the Timeline and modifying the parameterized dimensions (example shown for the first well; red rectangle). After resizing, the offset between the first well and the dish boundary should be updated to ensure the pattern remains within the circular mold footprint (red triangle).

(D) If resizing causes wells to overlap, spacing and/or the number of wells can be adjusted using the Rectangular Pattern Constraint (dotted black rectangle).

(E) Suggested stage-dependent dimensions to guide redesign: (E i) embryo measurements (dorso-ventral orientation) extracted from Kimmel et al. (1995); (E ii) corresponding recommended well parameters for FDM printing; (E iii) corresponding parameters for higher-precision printing (e.g., SLA). Right: schematic of the well geometry indicating the editable parameters A-D (mm) used throughout the tables.

Figure arrangement

The presence of FDM-printing-based mold design and vascularization in developing zebrafish embryos within the same figure may cause reader discomfort. Therefore, I recommend that the author separate the designing part and results into two distinct images.

We thank the reviewer for this suggestion and agree that combining the mold design workflow and the live imaging results within the same figure could be visually distracting. We have therefore followed the reviewer's recommendation and split these into two separate figures: the design and fabrication workflow in one figure and the imaging results in another (see revised Figures 1 and 2).

Figure 1. FDM-printed mold design.

A) CAD renderings of two circular mold sizes ($D/\varnothing = 20$ mm and 13 mm) with slot dimensions in millimeters. Each slot measures 0.40 mm wide at the narrowest point, 0.60 mm at the widest, and extends 5.50 mm deep; the spacing between slots is 1.00 mm, and the slot height above the

base is 2.00 mm. The smaller mold was designed for use with 14 mm coverslips, while the larger was tested with 21 mm coverslips. The mold design was kept circular rather than rectangular to fit tightly within the well and stay level. Design files for both sizes are available for download via the associated GitHub repository. B) Perspective 3D views of the assembled molds. Scale bar: 5 mm. C) Wells in 1.5% low-melting-point agarose were made using the seven- and twenty-tooth molds in 14- and 21-mm glass-bottom dishes, respectively. Scale bar: 10 mm. D) PrusaSlicer G-code preview illustrating how FDM resolution (0.4 mm nozzle) converts triangular cavities into rectangular slots. E) Exploded view of mold components, with a removable handle to facilitate positioning in small dishes.

Figure 2. FDM-printing-based mold design enables consistent live imaging of microglia motility and brain vascularization in developing zebrafish larvae.

A) Top left: image showing empty agarose wells created using the seven-teeth mold. Bottom left: image showing 56 hpf *Tg(fli1a:GFP-CAAX);Tg(gata1a:DsRed)* larvae inserted into the wells. The black-dotted box indicates the area magnified in the next panel. Right: magnified area of mounted larvae. Live brightfield imaging of the middle larvae, marked with an orange box, is presented in panel G. Scale bars: left panels 1 mm, right panel 500 μ m. B) Brightfield midplane-

volume image of the 56hpf *Tg(fli1a:GFP-CAAX);Tg(gata1a:DsRed)* larvae from panel F (orange boxed region), before the start of time-lapse acquisition, shown in panels C-D, orange box. Scale bar: 50 μm . C) Maximum intensity projection of horizontal sections from overnight time-lapse imaging of vasculature, *Tg(fli1a:GFP-CAAX)*, cyan), and erythrocytes, *Tg(gata1a:DsRed)*, red), in the developing brain of the mounted larvae (same larva as shown in panels A-B, orange box). The first timepoint is shown in panel (i). Temporal color coding of the *fli1:GFP* signal over the acquisition (time-lapse, 20-minute imaging interval, 32 time points) (ii). The color bar indicates the transition from time point 1 (magenta) to 32 (bright yellow). See also Movie S1. Scale bars: 50 μm . D) High-magnification (63x) imaging of the brain vascularization process. The maximum intensity projection of the first acquisition time point is shown in (i). The magnified area (indicated with an orange rectangle) shows filopodia-like protrusions extending from the *fli1:GFP*-positive (cyan) endothelial tip cell (ii). The *gata1a:DsRed* positive erythrocytes (red) inside a more mature blood vessel are also shown. See also Movie S1. Scale bars: 25 μm and 5 μm , respectively. E) Temporal color coding of microglia dynamics in *Tg(mpeg1.1:GFP-CAAX)* 72 hpf larvae, over the acquisition period (10-minute imaging interval, 80 time points). The color bar indicates the transition from time point 1 (magenta) to 80 (bright yellow). Scale bar: 50 μm . F) Selected time-points from panel E (blue box), showing microglia *Tg(mpeg1:GFP-CAAX)*, grey) migration and process elongation. See also Movie S2. Scale bar: 20 μm .

Quantitative assessment of positioning reproducibility

While qualitative images demonstrate consistent orientation, a quantitative metric (e.g., angle variance of the head relative to the coverslip across embryos) would strengthen the claim of improved throughput and consistency. If not, it is better to discuss why you couldn't do the quantitative analysis (in the "Discussion").

We thank the reviewer for this suggestion and agree that a quantitative measure would strengthen the claim of reproducible positioning. We therefore added a simple symmetry-based metric to quantify dorsoventral orientation consistency across a larger dataset. Specifically, we mounted 52 hpf wildtype larvae on one small and four large molds (71 larvae total) and acquired upright brightfield images with a Zeiss AxioZOOM.V16 stereo microscope at 16x magnification. We then quantified left-right similarity for the eye area using the score.

$$S = \frac{\min(L,R)}{\max(L,R)}$$

Where indicates identical values. High similarity suggests that both eyes are imaged oriented approximately symmetrically relative to the imaging plane; conversely, low similarity would indicate tilt or partial occlusion. Across the dataset, similarity values stay within a narrow range (area $S = 0.91 \pm 0.06$), supporting consistent positioning. These data have now been added as a new Supplementary Figure 1 (Fig. S1), and the related description has been included in the Methods section.

Figure S1. Quantitative assessment of positioning reproducibility using the left-right eye similarity ratio.

A) Representative brightfield images of 52 hpf larvae mounted in the FDM-based molds. (i) Left and right eyes are outlined (yellow) to illustrate the measurements used for quantification; the black-dotted box indicates the region shown at higher magnification (ii). Scale bars: 10 mm and 2 mm, respectively.

B) Distribution of left-right similarity ratio scores for eye area across all mounted larvae (n = 71). Similarity was calculated as $S = \min(L, R)/\max(L, R)$, where $S = 1$ indicates identical values between the left and right eyes.

Long-term viability assay

The manuscript mentions that larvae develop normally after release from agarose, but no data are shown (e.g., survival rate, heart rate, or behavioral assay after imaging). Providing at least a short viability assessment would reassure readers that the low agarose concentration and resin coating do not adversely affect development.

We thank the reviewer for requesting this helpful overview of mold utility in live imaging. Physical confinement in agarose is generally the most significant stressor for developing larvae, and we have not observed any mortality upon release from agarose. We have revised the text accordingly by removing the sentence about releasing from agarose and adding comments on larval viability when mounted in the molds. In the results (lines 113-114), we state:

“At both 54 hpf and 72 hpf, the mounted larvae exhibited normal development during imaging and maintained a healthy heartbeat at the end of the acquisition.”

In the discussion (lines 174-176), we write:

“Additionally, in our experience, larvae confined in agarose molds do not show reduced viability for at least 24 hours after mounting.”

Additionally, we evaluated the viability of larvae mounted at 52 hpf from a total of 5 zebrafish molds (A= small mold; B-E= large molds) 24 hours after mounting, keeping the molds at 28.5 °C in an incubator. The presence of a strong heartbeat was used to determine if the larvae were alive. As shown in the table below, all larvae were viable, indicating that, at least for an overnight imaging period, confinement in the agarose molds does not negatively impact larval development.

24h_post_mounting

Plate	Total (larvae)	Alive	Dead	% (alive)
A	7	7	0	100
B	20	20	0	100
C	13	13	0	100
D	18	18	0	100
E	18	18	0	100

Statistical analysis of imaging quality

Figures showing signal-to-noise ratio (SNR) or resolution metrics for low- versus high-NA objectives are presented, but statistical tests (e.g., t-test) are not reported. Adding appropriate statistical analysis and p-values will substantiate the claim of “no significant loss of signal”.

We thank the reviewer for suggesting this. However, statistical analysis here would not be appropriate, as we are not comparing SNR between low- and high-NA objectives. We are providing the SNR for indicative purposes only.

Application of the current technique

If the authors can provide additional results that suggest the current method can be applied to

© 2026. Published by The Company of Biologists under the terms of the Creative Commons Attribution License (<https://creativecommons.org/licenses/by/4.0/>).

study neuronal development, regeneration, and other related processes, it would enhance the manuscript's impact. Therefore, if it is feasible to perform ablation of a specific neuronal population and capture time-lapse images of neuronal regeneration or fin clipping in transgenic fish, followed by imaging of regeneration, such data would be highly valuable.

We agree with the reviewer that additional processes documented using the zebrafish molds designed here would enhance the scope and applicability of the manuscript. To demonstrate the feasibility of the mold design for imaging different cell types in the brain, we imaged microglial motility in 72 hpf *Tg(mpeg1.1:GFP-CAAX)* larvae (Fig. 2E). Similar to vascular sprouting, we captured the extension of dynamic microglial processes surveilling the surrounding tissue (Fig. 2F).

Figure 2E and 2F E) Temporal color coding of microglia dynamics in *Tg(mpeg1.1:GFP-CAAX)* 72 hpf larvae, over the acquisition period (10-minute imaging interval, 80 time points). The color bar indicates the transition from time point 1 (magenta) to 80 (bright yellow). Scale bar: 50 μm . **F)** Selected time-points from panel E (blue box), showing microglia *Tg(mpeg1:GFP-CAAX)*, grey) migration and process elongation. See also Movie S2. Scale bar: 20 μm .

Minor Points

Figure 1 B-C

Include scale bars for the printed molds.

Ensure gene/transgenic line names are italicized (e.g., *Tg(fli1a:GFP)*).

Done

References

Verify that all cited works (e.g., Geng & Peterson 2021) are present in the bibliography and follow the journal's style.

Done

Second decision letter

MS ID#: bio.062406R1

MS Title: Customizable FDM-based zebrafish larvae mold for live imaging

Authors: Marcela Xiomara Rivera Pineda, Jaakko Lehtimäki and Guillaume Jacquemet

I have now reached a decision on the above manuscript.

I greatly appreciate you taking the time to thoroughly address the reviewer comments. After reading your response to reviewer comments and the revised manuscript carefully, there are only a few small clarifications required. If you can make these, then we can publish your manuscript (without sending out for additional peer review).

Figure S1B - please define the graph clearly. If it is a box plot, what does the center line represent (mean? median?), what do the upper and lower bounds of the box represent? what do the error bars represent? what do the dots represent? Alternatively, you can show individual data points (though with so many data points $n=71$, I understand why you may have chosen to represent the graph as a box plot).

The response to reviewer 2 comments regarding 'Long-term viability assay.' The revised manuscript states "At both 54 hpf and 72 hpf, the mounted larvae exhibited normal development during imaging and maintained a healthy heartbeat at the end of the acquisition" and "Additionally, in our experience, larvae confined in agarose molds do not show reduced viability for at least 24 hours after mounting." The response to reviewer comments includes a table with data reporting viability, 24 h post mounting, for plates A-E. Can this please be included in the revised manuscript (as a table or supplemental table) in support of the statement above (which then may need to be moved from the Discussion to the Results section). It would be a shame to keep this valuable data buried in the reviewer comments and not share it with the world in your manuscript.

I should be grateful if you would also provide a point-by-point response detailing how you have dealt with the points raised by the reviewers in the 'Response to Reviewers' box. Please attend to all of the reviewers' comments. If you do not agree with any of their criticisms or suggestions please explain clearly why this is so.

Second revision

Author response to reviewers' comments

Figure S1B - please define the graph clearly. If it is a box plot, what does the center line represent (mean? median?), what do the upper and lower bounds of the box represent? what do the error bars represent? what do the dots represent? Alternatively, you can show individual data points (though with so many data points $n=71$, I understand why you may have chosen to represent the graph as a box plot).

Done. In the legend we now write: "Results are displayed as a box plot showing the median with the center line. The lower and upper edges of the box represent the 25th and 75th percentiles. The whiskers indicate the 5th and 95th percentiles, with individual dots representing data points outside this range."

The response to reviewer 2 comments regarding 'Long-term viability assay.' The revised manuscript states "At both 54 hpf and 72 hpf, the mounted larvae exhibited normal development during imaging and maintained a healthy heartbeat at the end of the acquisition" and "Additionally, in our experience, larvae confined in agarose molds do not show reduced viability for at least 24 hours after mounting." The response to reviewer comments includes a table with data reporting viability, 24 h post mounting, for plates A-E. Can this please be included in the revised manuscript (as a table or supplemental table) in support of the statement above (which then may need to be moved from the Discussion to the Results section). It would be a shame to keep this valuable data buried in the reviewer comments and not share it with the world in your manuscript.

Done. The table was added as Table S1.

Third decision letter

MS ID#: bio.062406R2

MS Title: Customizable FDM-based zebrafish larvae mold for live imaging

Authors: Marcela Xiomara Rivera Pineda, Jaakko Lehtimäki and Guillaume Jacquemet

Thanks for making the requested changes to Figure S1B and adding the table showing 24 hr viability. I am happy to tell you that your manuscript has been accepted for publication in Biology Open, pending our standard publication integrity checks. It was accepted on 4th March 2026.